# Phylogeny analysis of whole protein-coding genes in metagenomic data detected an environmental gradient for the microbiota

Soichirou Satoh[1,2]*, Rei Tanaka[2], Makio Yokono[3], Daiji Endoh[4], Tetsuo Yabuki[5], Ayumi Tanaka[6]

**1** Graduate School of Life and Environmental Sciences, Kyoto Prefectural University, Kyoto, Japan, **2** Faculty of Life and Environmental Sciences, Kyoto Prefectural University, Kyoto, Japan, **3** Division of Environmental Photobiology, National Institute for Basic Biology, Okazaki, Japan, **4** Department of Radiation Biology, School of Veterinary Medicine, Rakuno Gakuen University, Ebetsu, Japan, **5** General Education Department, Hokusei Gakuen University, Sapporo, Japan, **6** Institute of Low Temperature Science, Hokkaido University, Sapporo, Japan

* s-satoh@kpu.ac.jp

## Abstract

Environmental factors affect the growth of microorganisms and therefore alter the composition of microbiota. Correlative analysis of the relationship between metagenomic composition and the environmental gradient can help elucidate key environmental factors and establishment principles for microbial communities. However, a reasonable method to quantitatively compare whole metagenomic data and identify the primary environmental factors for the establishment of microbiota has not been reported so far. In this study, we developed a method to compare whole proteomes deduced from metagenomic shotgun sequencing data, and quantitatively display their phylogenetic relationships as metagenomic trees. We called this method Metagenomic Phylogeny by Average Sequence Similarity (MPASS). We also compared one of the metagenomic trees with dendrograms of environmental factors using a comparison tool for phylogenetic trees. The MPASS method correctly constructed metagenomic trees of simulated metagenomes and soil and water samples. The topology of the metagenomic tree of samples from the Kirishima hot springs area in Japan was highly similarity to that of the dendrograms based on previously reported environmental factors for this area. The topology of the metagenomic tree also reflected the dynamics of microbiota at the taxonomic and functional levels. Our results strongly suggest that MPASS can successfully classify metagenomic shotgun sequencing data based on the similarity of whole protein-coding sequences, and will be useful for the identification of principal environmental factors for the establishment of microbial communities. Custom Perl script for the MPASS pipeline is available at https://github.com/s0sat/MPASS.

**Data Availability Statement:** All relevant data are within the paper and its Supporting Information files. Perl script, which used in this study is publicly

available from the GitHub repository (https://github.com/s0sat/MPASS).

**Funding:** This study was supported by a grant from the Nippon Life Insurance Foundation (Environment, 2021-2022) to Soichirou Satoh, the Academic Contribution to the Region (ACTR) at Kyoto Prefectural University to Soichirou Satoh, and the Advanced Innovation powered by Mathmatics Platform (AIMaP) to Tetsuo Yabuki. The funders had no role in study design, data collection and analysis, decision to publish, or preparation of the manuscript.

**Competing interests:** The authors have declared that no competing interests exist.

## Introduction

The growth of microorganisms in nature is influenced by environmental factors, such as temperature, pH, carbon and nitrogen sources, and trace metal elements, and altered along with their gradient. Thus, correlation analysis of the transition between the composition of the microbiota and their environmental conditions (e.g., soil or water quality) can help to elucidate the establishment principles of microbial communities. Isolation and analysis of 16S rRNA sequences by denaturing gradient gel electrophoresis [1], which has been widely used for taxonomic composition analysis of microorganisms, has been largely replaced by amplicon sequencing using next-generation sequencers [2, 3]. Furthermore, the improved performance of sequencing technologies, the reduced cost of sequencing, and the development of bioinformatic methods have led to the development of whole metagenomic shotgun sequencing for the analysis of complicated microbiota [4–7]. Whole sequencing data include not only taxon-relating genes but also other functional genes that are involved in biological processes such as metabolism, signal transduction, and transport of ions and other chemicals. Therefore, metagenomic shotgun sequencing analysis can detect unique metabolic pathways, new species, and novel functional genes in addition to taxonomic identification. These multidimensional data have been integrated into publicly available databases such as KEGG (Kyoto Encyclopedia of Genes and Genomes) [8], MAPLE (microbiome analysis pipeline) [9], and KMAP (KAUST metagenomic analysis platform) [10] for functional analysis.

Although shotgun metagenomic sequence data are informative and make it possible to classify microorganisms with high taxonomic resolution, the data are complicated and methods for quantitative comparisons among whole metagenomic data are still being developed. For example, in the taxonomic assignment-based method proposed by Mitra *et al.* [11] the distance matrix of metagenomes is reconstructed based on the taxonomic profiles and ecological indices. The MEGAN Community Edition offers a wide variety of analysis options and can classify metagenomes based on taxonomic and functional assignments [12]. The Community-Analyzer developed by Kuntal *et al.* [13] is based on the pair-wise correlation of abundance of taxonomic groups. PhyloPhlAn3 uses sequences of highly conserved core genes to infer metagenomic phylogeny [14]. These methods can provide the precise composition of taxonomic and functional groups of microorganisms in addition to the distance between metagenomes. However, taxonomy- and function-based distances are not always the same as genomic distance. Indeed, a recent comprehensive analysis of 13,174 public metagenomic datasets across 14 major habitats indicated that most species-level genes and protein families were rare and the rates of positive selection for species-specific genes were low [15].

Another quantitative comparison method is the assignment-free *k*-tuple method [16]. A *k*-tuple (also called *k*-word or *k*-mer) is a segment of consecutive nucleotide bases of length *k*. When two genomes are closely related, the relative *k*-tuple frequencies will be more similar than those between distantly related genomes [17]. Dissimilarity in *k*-tuple frequencies between metagenomes has been used as the metagenomic distance for comparing and clustering metagenomic samples, and sophisticated parameters for metagenomic dissimilarities, such as $d^S_2$ and $d^*_2$, have also been developed and used for clustering metagenomes [18–20]. However, although *k*-tuple frequencies are influenced by gene density and codon bias in genomes, these factors are not directly related to the function of the genes. Here, we propose a function-related and assignment-free method based on phylogenomic considerations. Phylogenomics can identify evolutionary trajectories of organisms by comparing their genomic information. Several phylogenomic methods have been proposed, including concatenation of many alignments, construction of a supertree (consensus tree) from phylogenetic trees of each gene, and construction of a distance matrix from the averaged similarity of whole genes between

genomes [21]. The concatenation-based and supertree methods use traditional molecular phylogenetic techniques (e.g., maximum likelihood and maximum parsimony) to reconstruct the trees. Although the averaged sequence similarity method for constructing a distance matrix cannot use such traditional techniques, all the genetic information in the genome can be used and tree construction is performed automatically [22–24]. Human decisions, such as trimming of sequence alignments and selection of genomes and genes, are not required.

In this study, we developed a novel method for comparing whole protein-coding genes between metagenomic data that we named Metagenomic Phylogeny by Average Sequence Similarity (MPASS) (Fig 1). The MPASS method is derived from the average sequence similarity method. We applied MPASS to analyze simulated, soil, and aquatic microbiota datasets. We also analyzed the relationship between the metagenomic tree of samples from the Kirishima hot springs area in Japan and dendrograms of environmental parameters for this area [25] using a comparison tool for phylogenetic trees.

## Results

### Metagenomic tree analysis detected group relationships among simulated metagenomic datasets

We applied MPASS (Fig 1) to analyze two simulated metagenomic datasets of five bacterial species: *Sulfolobus islandicus*, *Proteus mirabilis*, *Nitrosospira multiformis*, *Bacteroides fragilis*, and *Acidobacterium capsulatum* [26]. In the first simulation, we varied the original bacterial amounts so that the species with high values were lower and the species with low values were higher (Fig 2A) and all values in each original vector converged. Thirty simulated metagenomic data samples, which were generated by mixing randomly sampled short reads from the five bacterial species, were divided into three groups (G1–3) based on the original bacterial composition. The metagenomic tree of these samples was reconstructed by MPASS (Fig 2B). The constructed tree reflected the three groups of metagenomic samples. The relationship between branching order and composition of the five bacterial species in the metagenomic samples was then examined in more detail. Within each group, the samples with the original bacterial composition were branched at the end of the clusters, whereas the samples with the greatest changes in bacterial composition (i.e., G1-9, G1-10, G2-7, G2-8, G3-9, G3-10) were deeply branched off. Because these deeply branched samples had lost their specific composition, this branching pattern of the tree is reasonable.

In the second simulation, we generated another 30 metagenomic data samples with random addition of Gaussian noise (Fig 3A). In the metagenomic tree of these samples (Fig 3B), almost all the samples separated into the corresponding clusters; the exceptions were G1-2 and G2-10, which were clustered in G2 and G1, respectively. The bacterial compositions of these two metagenomes were significantly different from those of the other metagenomes in each group (Fig 3A). Therefore, the branching pattern of this tree was congruent with the simulated metagenomic data. The results of the two simulation analyses strongly suggest that MPASS can reflect similarities of bacterial compositions among metagenomes.

### Metagenomic tree analysis detected group relationships among soil metagenomes

We applied MPASS to analyze the soil metagenomic data that Song *et al* [20] used to validate their microbiome clustering method. The 16 metagenomic data samples were derived from distinct geographic locations and ecological biomes, including hot deserts, cold deserts, forests, prairie, and tundra [27]. The metagenomic tree of these samples clearly separated them into

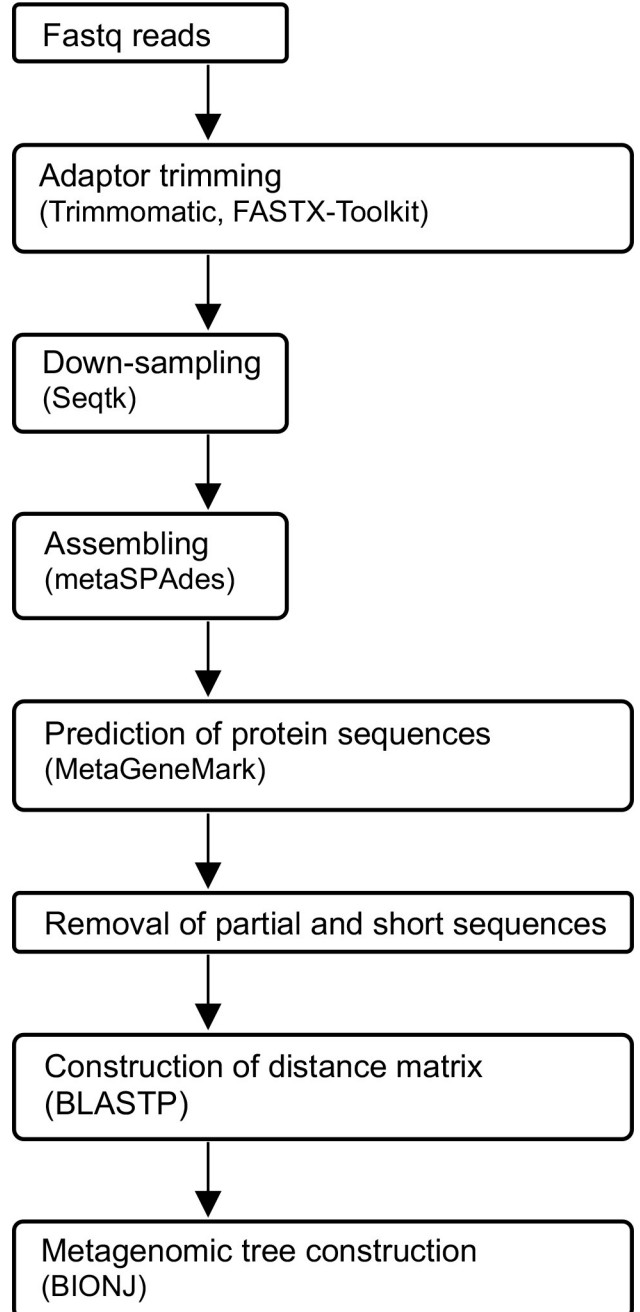

**Fig 1. Overall framework of the Metagenomic Phylogeny by Average Sequence Similarity (MPASS) method.**
Metagenomic fastq reads are assembled, and used to predict the protein-coding sequences. Down-sampling is performed to normalize the number of sequenced nucleotides. Incomplete and too short sequences are removed and the resultant proteome datasets are used to construct the distance matrix. Metagenomic trees are constructed using the neighbor-joining method. Quality filtering of fastq reads is optional, as appropriate.

cold desert, hot desert, and green biomes (forest, prairie, tundra) (Fig 4). Song *et al*. [20] found that the main factor for differentiation of soil microbiota was the soil pH value. For example, tropical forest (PE6) and Arctic tundra (TL1) samples were from soils with the lowest pH values (pH 4.12 and pH 4.58) among all the samples, and they clustered together in the tree. Our

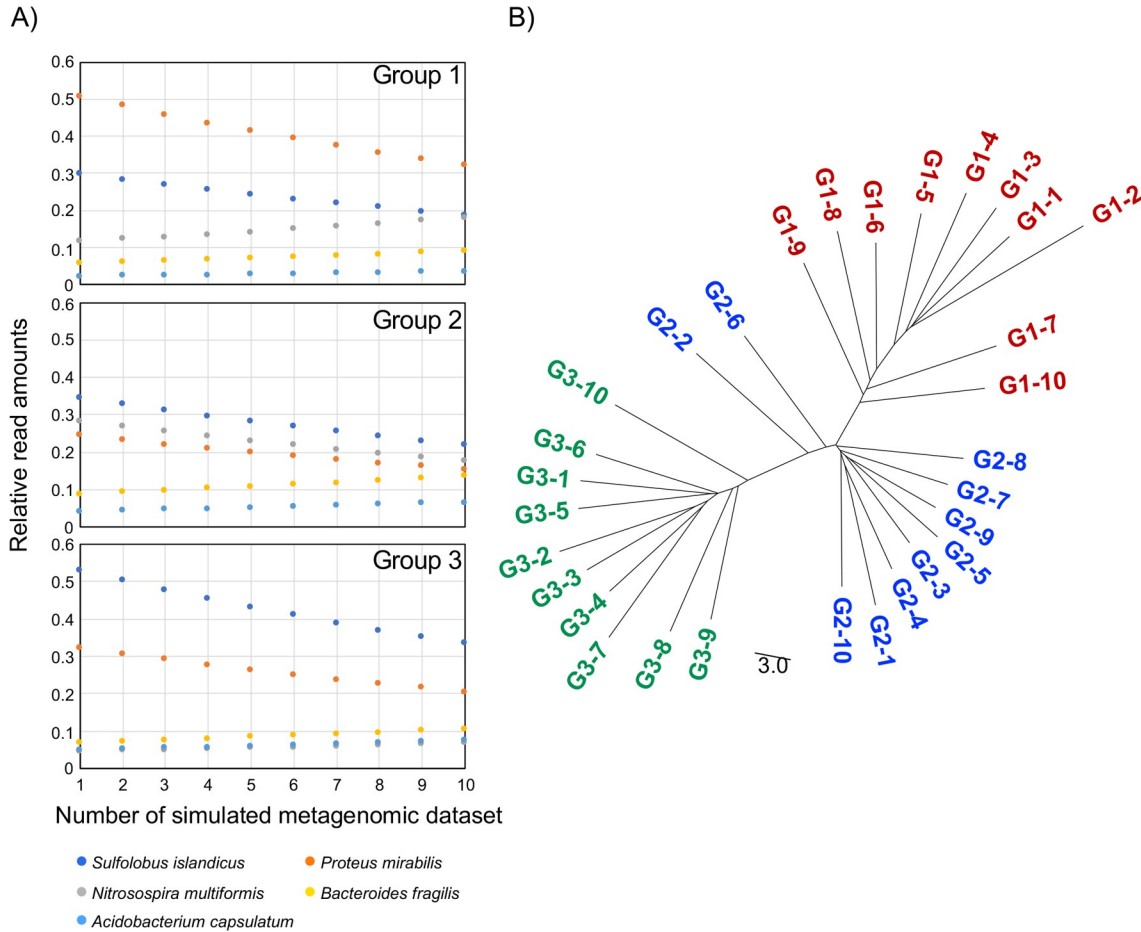

**Fig 2. Clustering of the first simulated metagenomic dataset.** Relative read amounts of five bacterial genomic sequences in the simulated metagenomic dataset (A). The horizontal axis indicates the sample number in the metagenomic tree; for example, sample 1 in Group 1 is G1-1 in the tree. Metagenomic tree based on the simulated metagenomes (B). The more complicated samples in each group, such as G1-9, G1-10, G2-7, G2-8, G3-9, and G3-10, are most deeply branched off from the corresponding clusters. Branch length indicates the nucleotide substitution rate as a percentage.

metagenomic tree also had subclusters within the three main clusters (cold desert, hot deserts, green biomes) that reflected similarities of soil pH values. The three metagenomic samples from soils with the lowest pH values, PE6, TL1, and boreal forest sample (BZ1), formed a subcluster in the main green biomes cluster. We found that geographic location also affected the clustering of the metagenomic samples; two cold (polar) desert samples (EB019 and EB026) and two hot desert samples (MD3 and SV1) from closely related locations formed subclusters in the two main clusters. These results indicate that MPASS can successfully cluster microbiota according to their environment and geographic location.

## Metagenomic tree analysis detected group relationships among aquatic metagenomes

We applied MPASS to analyze 35 aquatic metagenomic data samples derived from lakes, oceans, hot springs, and a deep-sea hydrothermal vent [25, 28–32]. The metagenomic tree of these samples clearly separated them into two main clusters, seawater and freshwater (Fig 5).

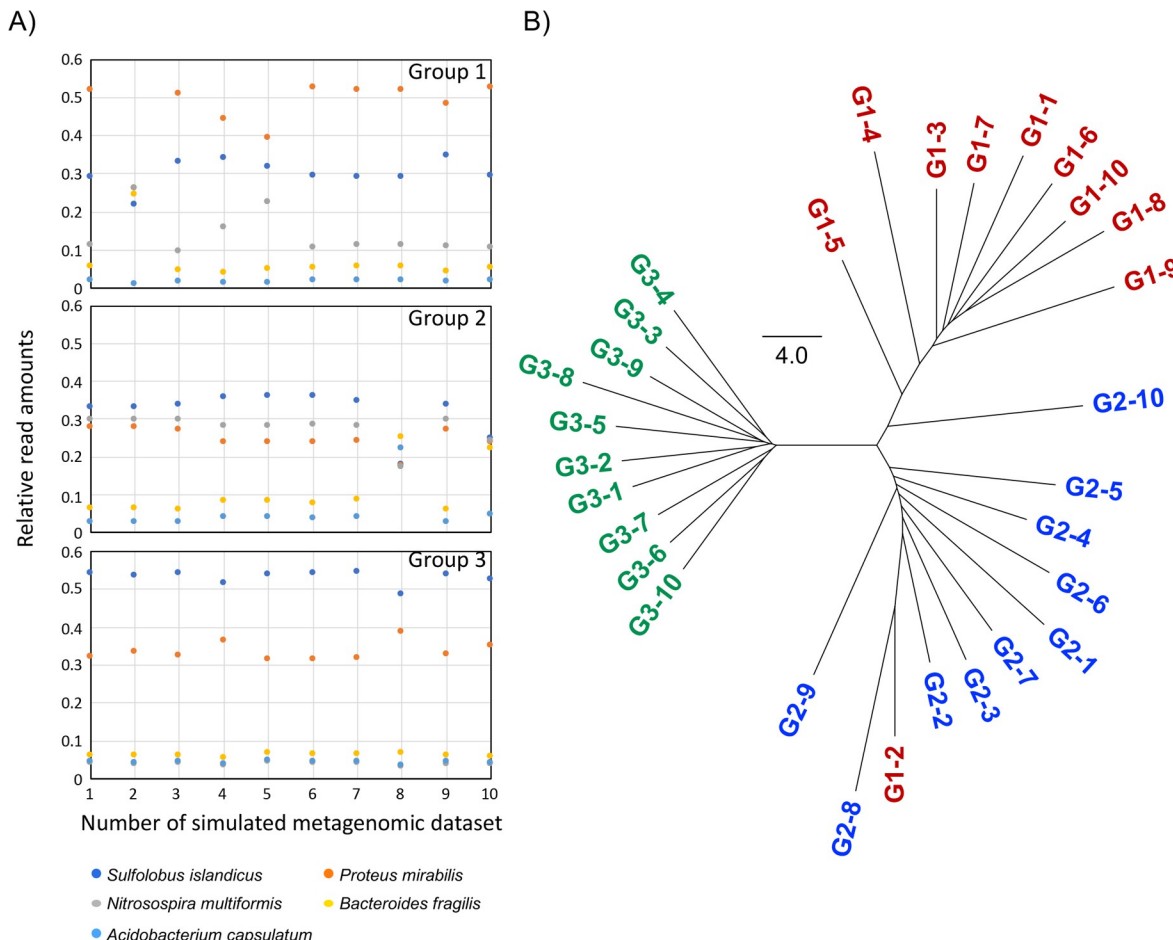

**Fig 3. Clustering of the second simulated metagenomic dataset.** Relative read amounts of five bacteria in the simulated metagenomic dataset with random addition of Gaussian noise (A). Metagenomic tree based on the simulated metagenomes (B). The relative read amounts of the five bacteria in the G1-2 and G2-10 samples are significantly different from those of other samples in each group, which may explain why G1-2 and G2-10 clustered in G2 and G1, respectively.

In the seawater cluster, the 15 samples formed three subclusters with distinct geographic locations (Fig 5). Among the three metagenomic samples from the *Tara* Ocean project [28], TarOc3 was the most deeply branched, and TarOc1 and TarOc2 were closely related. TarOc1 and TarOc2 were both derived from deep seawater (188 m), whereas TarOc3 was from surface water (5 m). The sampling sites of TarOc1 and TarOc2 also had lower temperatures and higher chlorophyll contents than the sampling site of TarOc3 (BioSample ID: SAMEA2622021 for TarOc1 and TarOc2, BioSample ID: SAMEA2621990 for TarOc3).

Among the four metagenomic samples from the Okhotsk Sea, WSP1 was most deeply branched, WSP5 was branched off, and WSP2 and WSP3 were closely related (Fig 5). This topology is consistent with a previously reported 16S rRNA-based clustering data analysis [31], which showed differences in the prokaryotic community composition; two groups of *Alphaproteobacteria* (*Rhodobacterales*) and one group of *Gammaproteobacteria* (*Oceanospirillales*) were more abundant in the WSP2, WSP3, and WSP5 samples than they were in the WSP1 sample.

Submarine hydrothermal systems contain hydrogen sulfide, methane, and hydrogen gasses and harbor chemolithoautotrophic bacteria and archaea [33, 34]. In the microbiota of a

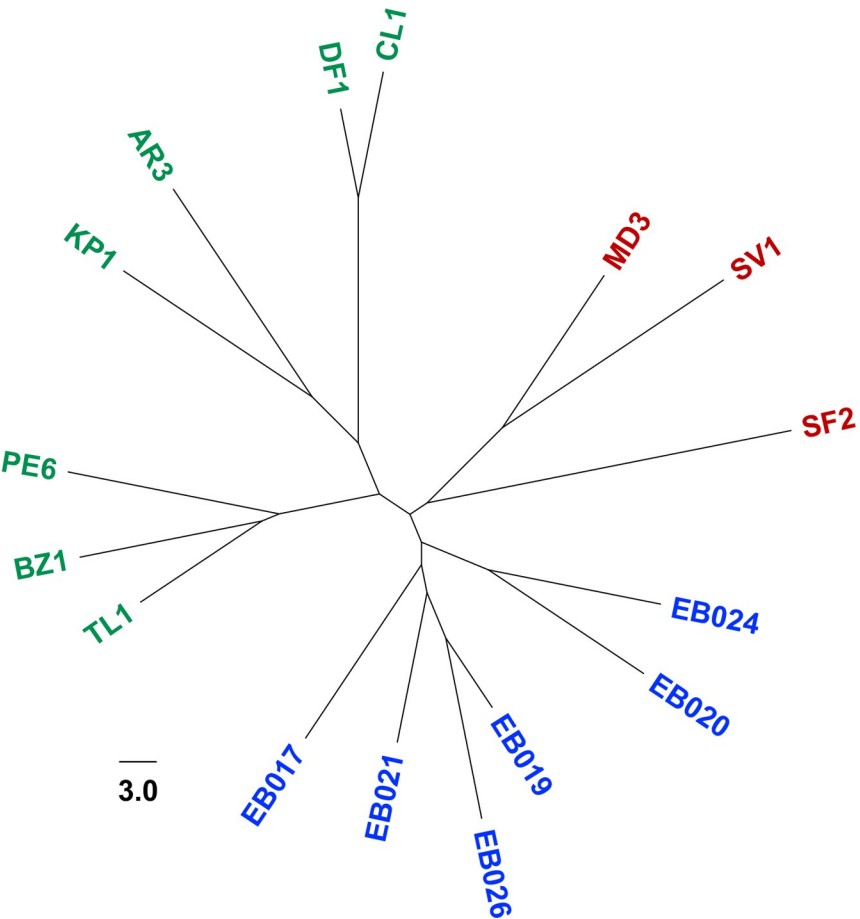

**Fig 4. Metagenomic tree for 16 soil samples from three ecologically distinct groups.** Red, hot desert samples; blue, cold (polar) desert samples; green, green biome samples. In the green biome subcluster, the pH of each sampling site was: AR3 (pH 5.90), BZ1 (pH 5.12), CL1 (pH 5.68), DF1 (pH 5.37), KP1 (pH 6.37), PE6 (pH 4.12), and TL1 (pH 4.58) [20].

hydrothermal vent around Kueishantao Island, *Epsilonproteobacteria* including *Campylobacteria*, which are classified as sulfur-reducing chemolithoautotrophic *Nautiliales*-like bacteria, was found to be a dominant group in the vent sites [30], whereas the abundance of *Epsilonproteobacteria* was much lower in the seawaters that were far from the hydrothermal vent. Our metagenomic tree showed that the two samples (W_outside and Y_outside) that were furthest from the hydrothermal vent formed a subcluster, but the other metagenomic samples did not cluster according to the distance from the hydrothermal vent (Fig 5). This may be because the locations from which the metagenomic samples were derived reflected dynamic changes in the atmospheric environment between the hydrothermal vent and its surroundings.

In the freshwater cluster, the 20 samples clearly separated into lake and hot spring subclusters (Fig 5). Although the metagenomic samples from the Kirishima hot springs area were not separated from the central India hot spring samples at the root of these branches, the Kirishima samples formed a single cluster. In the lake subculture, the four metagenomic samples from the boreal lake in Finland were taken from water layers at different depths; sari1b, sari2b, sari3b, and sari4b were from water layers at 0.1 m, 1.1 m, 1.6 m, and 2.1 m, respectively [32]. In this lake, although the oxygen concentration was constant from the surface water to a depth of 1 m, it rapidly decreased at a depth of 2 m. The methane and carbon dioxide concentrations

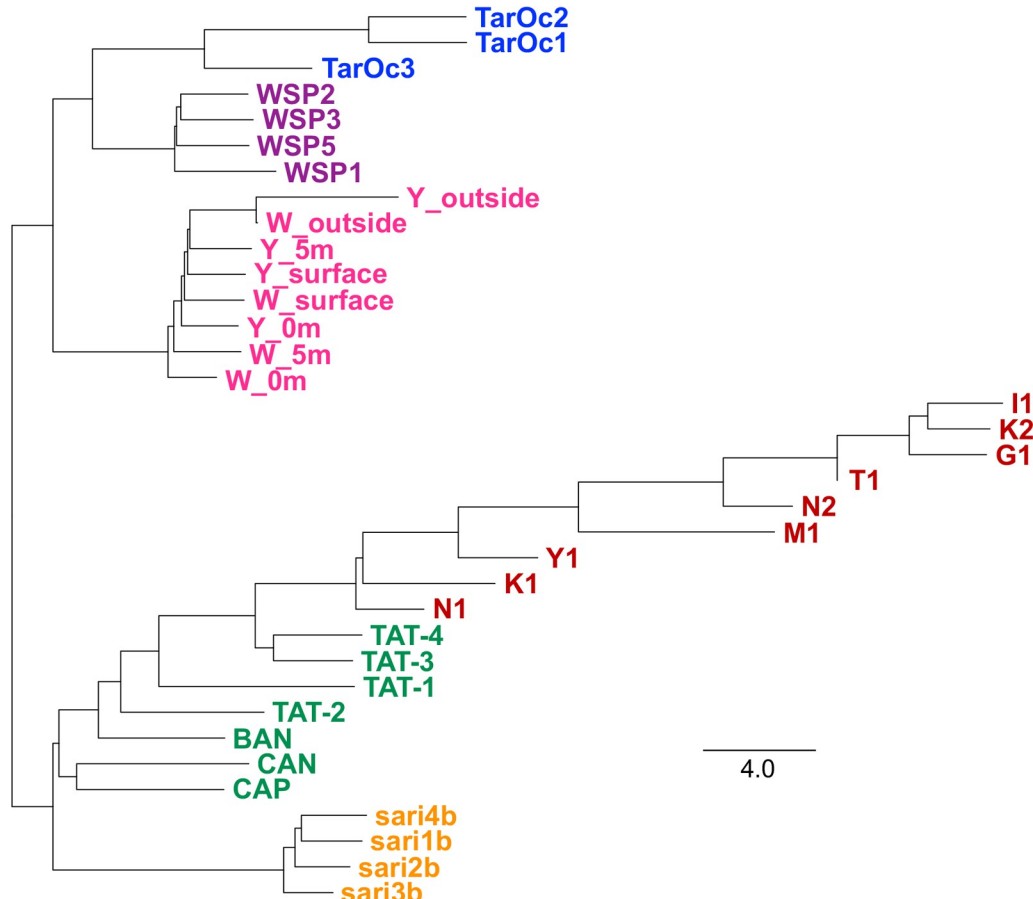

**Fig 5. Metagenomic tree for 35 aquatic samples from six ecologically distinct groups.** Blue, offshore samples; purple, samples from the coastal areas; magenta, samples from submarine hydrothermal vent; orange, lake samples; green and red, two hot spring samples. In the subcluster of the Kirishima hot spring samples (red), the temperature at each sampling site was: G1 (85.5°C), I1 (88.0°C), K1 (84.5°C), K2 (94.7°C), M1 (96.8°C), N1 (68.0°C), N2 (89.1°C), T1 (84.0°C), and Y1 (90.4°C) [25]. In the subcluster of central India hot spring samples (green), the temperature at each sampling site was: BAN (55.0°C), CAN (43.5°C), CAP (52.1°C), TAT-1 (98.0°C), TAT-2 (61.5°C), TAT-3 (67.0°C), and TAT-4 (69.0°C) [29].

were almost the same in all four layers. The branching pattern of these samples in our metagenomic tree did not reflect the depths of the sampling areas; sari3b was most deeply branched, sari2b was branched off, and sari1b and sari4b, were most closely related. This branching pattern may be caused by other environmental factors.

In the hot springs subcluster, all the samples from central India [29] except the TAT-1 sample, were branched off in order of temperature (Fig 5). Samples CAN (43.5°C) and CAP (52.1°C) from the lowest temperature sites were the most deeply branched, then BAN (55°C) and TAT-2 (61.5°C) were branched off. The remaining three metagenomic samples formed a subcluster in which TAT-1 (98°C) branched off first and TAT-3 (67°C) and TAT4 (69°C) were most closely related. The TAT-3 and TAT-4 samples had lower amounts of proteobacteria than the other metagenomic samples.

The branching pattern of the Kirishima samples in our metagenomic tree (Fig 5) correlated with the turbidity and pH of the sampling sites. The four deeply branched samples, N1, K1, Y1, and M1, were taken from the hot springs with the higher pH and lower turbidity [25], whereas the other five samples were from the hot springs with the lower pH and higher

turbidity. The N1 sample, which taken from the hot spring at 68˚C, was most deeply branched, then the K1, Y1, and M1 were branched off. *Sulfolobaceae*, which is in superkingdom Archaea, was included in the K2, I1, T1, G1, and N2 samples. Archaea in this family are known to be adapted to high temperature and low pH environments; therefore, it is likely that the branching pattern of the Kirishima subcluster is related to the pH, turbidity, and temperature of the sampling sites. Together, these results show that the branching patterns of these aquatic metagenome samples reflect their environmental factors, microbial communities, and geographic locations, and indicate the suitability of MPASS for the clustering of microbiota.

## Metagenomic tree topology was similar to that of dendrograms of highly influential environmental factors

Quantifying the relationship between metagenomic transitions and changes in environmental parameters, including geology and water quality, can indicate the extent of the influence of the environment on the microbiota. We used the branches for the Kirishima samples in our metagenomic tree (Fig 5) as a model case. We applied the TREEDIST program [35], which is used to compare the topologies of two phylogenetic trees, to quantitatively compare our metagenomic tree with dendrograms of previously reported environmental parameters. The environmental parameters for the Kirishima metagenomic samples reported by Nishiyama *et al.* [25] were as follows: M1, K2, I1, T1, G1, and N2 were derived from a hot spring with high turbidity, low pH, high concentration of sulfate, copper and zinc ions, and relatively high total organic carbon and total nitrogen, whereas K1, Y1, and N1 were derived from transparent and medium pH environments. These properties of hot springs are related to their microbiota; *Crenarchaeota* and *Aquificae* were dominant in the former and latter hot springs, respectively. We constructed 41 dendrograms based on the environmental parameters of the Kirishima samples.

Fig 6 shows the topological similarity between the metagenomic tree and the 41 dendrograms for the environmental parameters. The similarity was quantified as symmetric difference, which is based on the number of branching points with identical topology [36]. A low symmetric difference implies that two trees are very similar. Our metagenomic tree was similar to the dendrograms constructed based on oxidation-reduction potential, total nitrogen, total organic nitrogen, dissolved organic nitrogen, particulate organic nitrogen, and concentration of sulfate ($SO_4$) and vanadium ions. *Crenarchaeota* is aerobic and known to dominate in sulfate-rich hot springs [37, 38], and *Aquificae* is anaerobic [39, 40]. Oxidation-reduction potential is strongly related to oxygen concentration, and therefore the low symmetric differences for oxidation-reduction potential and $SO_4$ and vanadium ion concentrations suggest that these environmental parameters are strongly related to the establishment of microbiota in the Kirishima area.

## Metagenomic tree topology reflected the dynamics of microbiota at the organismal and genetic levels

In phylogenomic trees, closely related genomes share many orthologs and distantly related genomes share a small number of orthologs, including house-keeping genes, paralogs, and other genes. We analyzed our MPASS-based metagenomic tree to determine how many species and genes were distributed across the various metagenomes on the same lineage and how many genes were shared between closely related metagenomes. We choose 1,000 high-coverage protein sequences in the I1 and N1 metagenomes, which are at opposite ends of the Kirishima metagenomic subcluster in Fig 5, and calculated the existence probability of these proteins in the other Kirishima metagenomic samples. The evolutionary distances for each

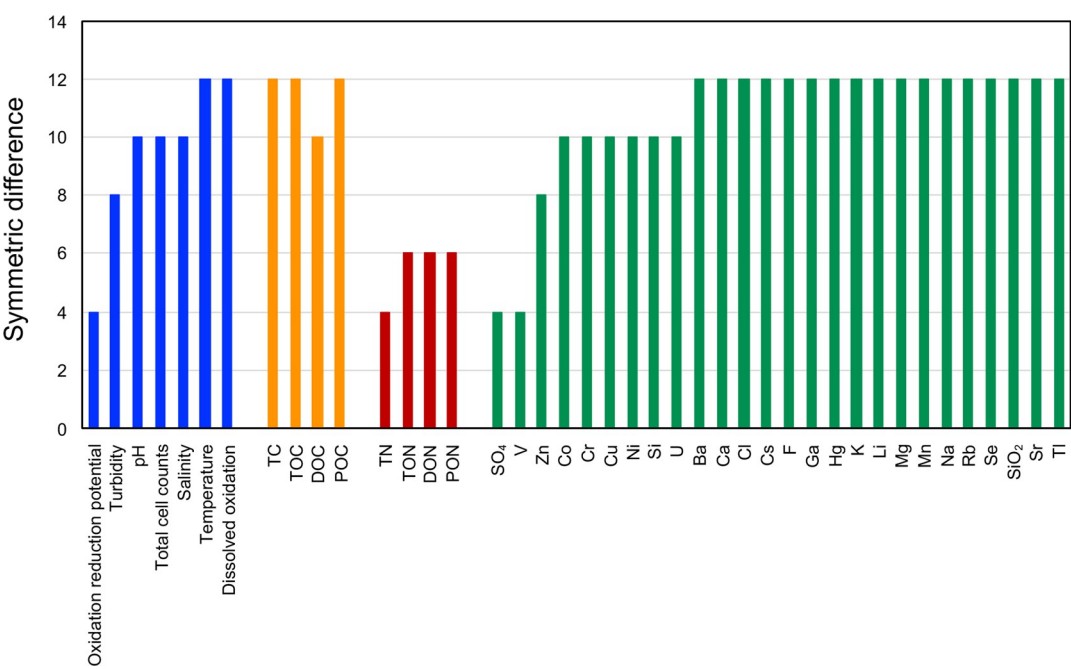

**Fig 6. Similarities between the metagenomic tree and environmental properties of Kirishima hot spring samples.** The vertical axis indicates the symmetric difference between the metagenomic tree and a dendrogram for each environmental property. A low symmetric difference indicates that the metagenomic tree and the dendrogram are similar. Blue, water quality; orange, carbon concentration; red, nitrogen concentration; green, metal and ion concentration. TC, total carbon; TOC, total organic carbon; DOC, dissolved organic carbon; POC, particulate organic carbon; TN, total nitrogen; TON, total organic nitrogen; DON, dissolved organic nitrogen; PON, particulate organic nitrogen.

gene were obtained using the same calculation procedures as were used in MPASS without averaging the gene distances in the metagenomic samples, and were used as the existence probability. The results are shown in the heatmaps in Fig 7A and 7B. As mentioned in the previous section, the composition of microorganisms and environmental properties of I1 were relatively similar to those of M1, K2, T1, G1, and N2 [25], whereas those of N1 were similar to those of K1 and Y1. Fig 7A shows that almost all of the homologous proteins of the I1 proteins existed only in the K2, G1, and T1 samples, and most of the homologous proteins of the N1 proteins existed mainly in the K1 and Y1 samples (Fig 7B). Unexpectedly, the N2 and Y1 samples contained many homologous proteins of both the I1 and N1 proteins. We selected 454 and 515 I1 and N1 proteins, respectively, that showed similar distribution patterns as the I1 and N1 samples in the MPASS-based metagenomic tree (Pearson correlation coefficient >0.7) for further analysis. These results indicate that approximately 50% of the 1,000 high-coverage protein sequences were distributed across the various metagenomes on the same lineage as the metagenomic tree.

The composition of microorganisms estimated from the selected I1 or N1 proteins is shown in Fig 7C and 7D. In the I1 protein group, three phyla of archaea, *Crenarchaeota*, *Nanoarchaeota*, and *Candidatus Thermoplasmatota*, and two groups of viruses were found (Fig 7C). The *Crenarchaeota* proteins were the most numerous and accounted for 71% of the I1 proteins. In the N1 protein group a variety of bacteria, archaea, and viruses was found (Fig 7D). The *Aquificae* proteins accounted for 91% of the N1 proteins. These results are consistent with the previously reported taxonomic composition of microorganisms based on 16S rRNA sequences [25], and strongly suggest that MPASS can successfully elucidate metagenomic dynamics among various microbiota.

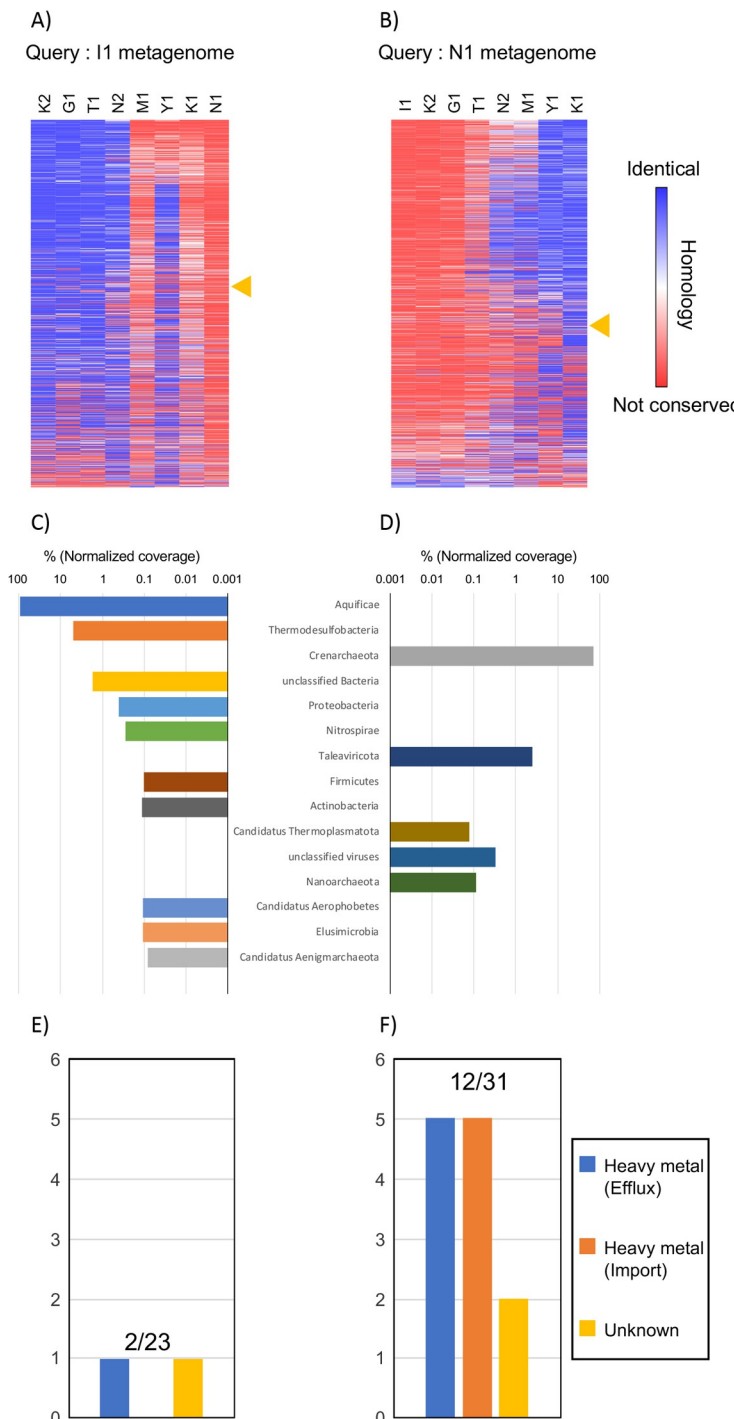

**Fig 7. Conservation of protein sequences among metagenomes.** Heatmaps indicate the conservation of 1,000 major protein sequences in the I1 and N1 metagenomes in other metagenomes (A, B). Protein sequences are sorted in descending order by the Pearson correlation coefficients between conservation in other metagenomes and the branch length in the metagenomic tree. Yellow triangles indicate the position at which the correlation coefficient is 0.7, the cut-off point. Taxonomic composition of protein sequences that had similar distribution patterns as the Kirishima hot spring subcluster in the metagenomic tree (C, D). Numbers of the transporter protein sequences with the same distribution patterns of the proteins in Fig 7C-7F. Numbers of metal transporter proteins (numerator) and all transporters (denominator) are indicated above the bar plots. Proteins for the light-metal transporters were not observed. Data for the I1 homologs (Fig 7A, 7C, 7E) and N1 homologs (Fig 7B, 7D, 7F).

As reported previously [25], the N1 protein group contained a NapA protein and two Sox proteins, SoxB and SoxZ (S4 Table). NapA is involved in nitrogen fixation and nitrification and SoxB and SoxZ are involved in the sulfur metabolism pathway. We also found differences in the types of transporter proteins between the I1 and N1 protein groups; metal transporters were frequently found in the N1 proteins (Fig 7F) but rarely found in the I1 proteins (Fig 7E). This result suggests that the N1 metagenome may consist of more diverged microorganisms that can use various metal elements that are imported from the environment by metal transporters. Low pH conditions, such as that for the N1 sample, can increase the solubility of metals [41]. Therefore, the ability to use metal ions might be needed for a microorganism to become dominant in the N1 microbiome.

In this study, we have shown that the gradients of both major bacterial species and environmental factors are closely related to the establishment of microbiota and their communities. We also discovered a new environmental factor for Kirishima microbiota; namely, the gradient of vanadium concentration, which was highly correlated with the topology of our metagenomic tree (Fig 6). Our results show that the MPASS method has the potential to discover new environmental factors and to detect their effect on microbiome formation.

## Discussion

Quantitative comparisons between metagenomes can detect influential environmental factors for microbiota, discover useful or harmful microorganisms, and define the interaction of microbiota with their environments. Such comparisons present difficult challenges because genetic and non-genetic sequences from numerous organisms are present in metagenomic sequence data, and their functions are also diverse. In this study, we focused on the protein-coding genes in metagenomes, compared the amino acid sequences of whole proteins among metagenomes, quantitatively compared the coverage of protein sequences with the similarity between metagenomes, and clustered metagenomic data as a phylogenetic tree. These processes were defined as the MPASS method, which we applied to analyze three metagenomic datasets of simulated, soil, and aquatic samples. We showed that MPASS accurately clustered each of these datasets.

Furthermore, because phylogenetic trees can be used to infer the evolution of organisms with their adaptation processes, we used our metagenomic tree to infer the transition of microbiota in response to environmental factors. We showed that the branching orders of the metagenomic samples in the hot springs and hydrothermal vent clusters correlated well with increases in both the temperature of the sampling sites and their distance from the hydrothermal sources (Fig 5). In the cluster of the samples from the *Tara* Ocean project, a sea surface-derived sample first branched off, and a subcluster of deep-sea samples was located at the end of our metagenomic tree. Thus, the MPASS-based metagenomic tree successfully represented transitions in microbiota that occurred concomitantly with the environmental gradients. The MPASS-based tree and TREEDIST program [35] quantified the relationships between the microbiota and their environmental factors as shown in Fig 6.

Other phylogenomic methods have been proposed [21], such as concatenation of many alignments and construction of a supertree (consensus tree) from many phylogenetic trees on each gene. Our MPASS method is based on the average sequence similarity method [21, 23, 24], which can reconstruct phylogenomic trees without the need for human decisions, such as trimming of alignments and selection of genomes and genes. We also used MPASS to compare the extremely different I1 and N1 metagenomes that do not share any genes as shown in Fig 7A and 7B. According to a previously reported 16S rRNA analysis, K2, I1, T1, and G1 metagenomic samples contain *Crenarchaeota* species but not *Aquificae* species [25]. In such a case,

concatenated alignments and supertrees cannot be reconstructed because construction of alignment and phylogenetic trees require a set of common genes. Our MPASS method can reconstruct a metagenomic tree, regardless of the number of shared genes between metagenomes. Furthermore, genes that exist in only one of two metagenomes can also be used to estimate metagenomic distances, which is important for the construction of accurate phylogenomic trees [24]. In this study, we integrated the coverage of each gene into our previous method [23] as a new parameter. The topology of the metagenomic trees of simulated metagenomic datasets (Figs 2 and 3), soil (Fig 4), and aquatic samples (Fig 5) detected reliable group relationships, however, the integration of the coverage of each gene into metagenomic distances will require further theoretical consideration in future studies.

Visualization of inter-microbiome interactions is also important for the study of microbial ecosystems. Principal coordinates analysis plots [42] and phylogenetic networks [43] have been used to represent one-to-many and many-to-many interactions between microbiota. We used the classical binary tree-like representation method because, although this method is restrictive, sophisticated phylogenetic techniques can be used to analyze the resultant tree-like structures. We also used the TREEDIST program [35] to compare the branching patterns between our metagenomic tree and dendrograms that were reconstructed with environmental factors, and successfully elucidated influential environmental factors for the microbiota (Fig 6). The best choice of methods may be solved by the introduction of comparison methods for the topology and branching pattern of network-like graphs.

In this study, we showed that the inventory of protein-coding genes was a useful character for comparisons between microbiomes and their environment. Although MPASS can use datasets that are larger than those of 16S amplicon-based methods, reduced memory usage and improved processing speed can be achieved by replacing the bioinformatics tools; for example, metaSPAdes [44] and BLASTP [45] can be replaced with MEGAHIT [46] and MMseqs2 [47], respectively. Furthermore, MetaPlatanus [48] is useful for the assembly of long reads or mate-paired reads, and therefore, long-read and cost-effective sequencing technology using Nanopore sequencers will fit our method better and make MPASS easier to use.

## Materials and methods

### MPASS method

The MPASS method mainly includes the following steps: down-sampling, and assembly of Illumina short reads; prediction and translation of protein-coding genes; removal of partial and short sequences; construction of the distance matrix based on whole proteome comparisons; and reconstruction of the phylogenomic tree of metagenomes. Quality filtering of short reads is optional, as appropriate. The overall framework of the MPASS method is given in Fig 1 and details of each step are given below. Custom Perl script for this method is available at https://github.com/s0sat/MPASS.

### Quality filtering, down-sampling, and assembly of short sequence reads

Raw Illumina short reads were adapter-trimmed using Trimmomatic with the following parameters: ILLUMINACLIP:2:30:10 LEADING:20 TRAILING:20 SLIDINGWINDOW:4:15 MINLEN:50 [49]. Reads, which was sequenced using a MiSeq Reagent Kit v2, were trimmed to 200 bp using the FASTX-Toolkit (http://hannonlab.cshl.edu/fastx_toolkit). Prior to assembly, random sampling of reads was performed using seqtk (https://github.com/lh3/seqtk) to obtain the lowest number of reads among the metagenomic data used in the tree reconstruction. The reads were assembled using metaSPAdes [44].

## Prediction and translation of protein-coding genes, and removal of partial and short sequences

Protein-coding genes in the obtained scaffolds were predicted and translated into amino acid sequences using MetaGeneMark [50]. Partial amino acid sequences with no start and stop codons in the encoding nucleotide sequences were removed by custom Perl script. Brocchieri and Karlin [51] and Tiessen *et al.* [52] found that very short sequences contained many artifacts from sequence assembly, and therefore amino acid sequences with <100 amino acids were also removed by custom Perl script.

## Construction of the distance matrix based on whole proteome comparisons

Construction of the distance matrix by comparisons between proteomes was performed according to Satoh *et al.* [23] with two modifications. First, we integrated the coverage of contigs into the calculation of proteomic distances. Unlike gene sequences in genomic data, metagenomic data contain redundant gene sequences, which reflects the relative numbers of microorganisms in the samples. The *k*-tuple coverage of each protein-coding gene was obtained using metaSPAdes in the assembling step, and used to weight the protein sequences (see S1 Text for details). Next, we estimated the new constants (const1 and const2) to obtain the distances between genome data S as follows:

$$S = const1 \times e^{(const2 \times T)},$$

where T is the normalized average similarity between proteomes. In our previous report [23], these two constants were obtained from a comparative analysis of 55 prokaryote genomes and their 16S rRNA sequences. We recalculated these constants from the genomic data of a balanced selection of 38 prokaryotes, 25 eukaryotes, and 31 archaea according to Yokono *et al.* [24] (S1 Table). The relationship between distances of whole protein sequences between organisms and 16S rRNA sequence substitution rates is shown in S1 Fig. The 16S rRNA nucleotide sequences were obtained from the SILVA NR99 database (v138.1) [53]. Whole protein sequence data of each organism were downloaded from GenBank. From the distribution of the points in the plot of genomic distances against the percentage of Poisson-corrected substitution rates of 16S rRNA genes, const and const2 were estimated as 4.142 and 2.824, respectively (S1 Fig). The Poisson-corrected substitution rates were used to correct multiple substitutions in the same nucleotide sites of 16S rRNA genes [54]. The percentage of Poisson-corrected substitution rate ($S_2$') was calculated as follows:

$$S_1{}' = 1 - \frac{matched\ loci}{alignment\ length - gapped\ loci}.$$

$$S_2{}' = -\ln\left(1 - S_1{}'\right) \times 100.$$

The names of the organisms that were used to estimate these two constants are listed in S1 Table.

## Preparation of two simulated metagenomic datasets

Simulated short-read metagenomic datasets were obtained using NeSSM software [55] and the genomic sequences of five bacterial species, *Sulfolobus islandicus*, *Proteus mirabilis*, *Nitrosospira multiformis*, *Bacteroides fragilis*, and *Acidobacterium capsulatum* [26]. For the first simulation, we prepared three relative abundance vectors of these species (0.297, 0.507, 0.116, 0.058, 0.022), (0.345, 0.244, 0.281, 0.088, 0.042), and (0.526, 0.320, 0.042, 0.066, 0.046) according to

the simulation method of Jiang *et al.* [19]. From these abundance vectors, we generated 30 vectors in which each of the five values was increase or decrease by 5%, and used them to generated 30 metagenomic samples by mixing the randomly sampled short reads from the five bacteria using NeSSM [55].

In the second simulation, we generated 30 vectors from the original species abundance vectors used for the first simulation. These 30 vectors were generated by adding to each component the absolute value of one-fifth Gaussian noise, with mean zero and standard deviation equal to the value of that component. Each species abundance vector was randomized and renormalized 10 times, and the 30 vectors, which belonged to three groups with 10 vectors in each group, were obtained (Fig 2A). These vectors were used to generate 30 metagenomic samples as was done in the first simulation.

## Soil metagenomic dataset

We used 16 soil metagenomic datasets from various sites and biota from Song *et al.* [20]: hot deserts, cold (polar) deserts, and green biomes (forests, prairie grassland, tundra). Sequence data were downloaded from the MG-RAST server [56]. The accession numbers for the downloaded sequences are given in S2 Table.

## Aquatic metagenomic and environmental datasets

The aquatic metagenomic dataset includes 35 microbiome samples from fresh and seawater, hot springs, and a hydrothermal vent [25, 28–32]. Sequence data were downloaded from the Sequence Read Archive (SRA) and DDBJ Read Archive (DRA) databases. The accession numbers for the downloaded sequences are given in S3 Table. The environmental dataset for the Kirishima hot spring area are from Nishiyama *et al.* [25]. This dataset was used for the correlation analysis between the metagenomic tree and gradients of environmental factors.

## Quantification of similarity between metagenomic tree topology and gradients of various environmental parameters

The similarity of tree topologies between the metagenomic tree and the dendrograms based on environmental factors reported by Nishiyama *et al.* [25], including pH, temperature, turbidity, and concentrations of carbon, nitrogen, and various ions, was determined using the TREE-DIST program [35]. The similarity between the tree and dendrograms was quantified as symmetric difference, which is based on the number of branching points with identical topology [36].

## Correlation analysis between conservation of each gene and the topology of the metagenomic tree

Conservation of high-coverage protein sequences in the I1 and N1 metagenomic samples in other metagenomic samples was estimated based on the metagenomic distance S, which was calculated using the formula given above. We used the e-values of each protein sequence instead of T, the normalized average similarity of proteomes. To avoid using unreliable putative proteins in this analysis, we chose the top 1,000 proteins with high *k*-tuple coverage from the I1 and N1 samples. Then, we calculated Pearson correlation coefficients for each protein between the metagenomic distance S and branch length in the MPASS-based tree for the aquatic microbiota. Distance S of each protein in the I1 and N1 samples was sorted with the corresponding Pearson correlation coefficient, and plotted as heatmaps using the Morpheus web-based tool (https://software.broadinstitute.org/morpheus/).

Proteins with correlation coefficients >0.7 were taxonomically and functionally annotated by BLASTP searches against the nr-aa (non-redundant amino acid sequence) database with an e-value cutoff of $10^{-20}$. Species names and protein functions of the top hit were used for the annotation.

## Supporting information

**S1 Text. Procedure for estimating the metagenomic distance in the MPASS method.**
(DOCX)

**S1 Fig. Regression curve of genomic distances against substitution rates of 16S rRNA genes.**
(TIF)

**S1 Table. List of organisms used for the regression analysis.**
(XLSX)

**S2 Table. MG-RAST sample IDs of soil metagenomic data.**
(XLSX)

**S3 Table. SRA accession numbers of water metagenomic data.**
(XLSX)

**S4 Table. List of N1 group proteins related to nitrogen and sulfur metabolism.**
(XLSX)

**S1 File. Original data for all metagenomic trees and their related data.**
(ZIP)

## Acknowledgments

We thank Dr. Gento Tsuji and Dr. Hideaki Miyashita for helpful discussions. We also thank Margaret Biswas, PhD, from Edanz (https://jp.edanz.com/ac) for editing a draft of this manuscript.

## Author Contributions

**Conceptualization:** Soichirou Satoh, Makio Yokono, Daiji Endoh, Tetsuo Yabuki, Ayumi Tanaka.

**Data curation:** Soichirou Satoh, Rei Tanaka.

**Formal analysis:** Soichirou Satoh, Rei Tanaka.

**Funding acquisition:** Soichirou Satoh, Tetsuo Yabuki.

**Investigation:** Soichirou Satoh, Rei Tanaka.

**Methodology:** Soichirou Satoh, Rei Tanaka, Makio Yokono, Daiji Endoh, Ayumi Tanaka.

**Project administration:** Soichirou Satoh.

**Software:** Soichirou Satoh, Makio Yokono.

**Supervision:** Soichirou Satoh.

**Validation:** Soichirou Satoh.

**Writing – original draft:** Soichirou Satoh, Rei Tanaka, Ayumi Tanaka.

**Writing – review & editing:** Soichirou Satoh, Makio Yokono, Daiji Endoh, Tetsuo Yabuki, Ayumi Tanaka.

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
