## [Decision Letter · Decision Letter 0]

13 Oct 2022

PONE-D-22-20665Phylogeny analysis of whole protein-coding genes in metagenomic data detected an environmental gradient for the microbiotaPLOS ONE

Dear Dr. Satoh,

Thank you for submitting your manuscript to PLOS ONE. After careful consideration, we feel that it has merit but does not fully meet PLOS ONE’s publication criteria as it currently stands. Therefore, we invite you to submit a revised version of the manuscript that addresses the points raised during the review process.

As you can see the two reviewers gave completely opposite answers, I found your paper potentially interesting, however, the important technical flaws highlighted by one of the reviewers should be addressed.==============================

We look forward to receiving your revised manuscript.

Kind regards,

Alfonso Esposito, PhD

Academic Editor

PLOS ONE

Journal Requirements:

Upon re-submitting your revised manuscript, please upload your study’s minimal underlying data set as either Supporting Information files or to a stable, public repository and include the relevant URLs, DOIs, or accession numbers within your revised cover letter. For a list of acceptable repositories, please see http://journals.plos.org/plosone/s/data-availability#loc-recommended-repositories  Any potentially identifying patient information must be fully anonymized.

4. We note that you have stated that you will provide repository information for your data at acceptance. Should your manuscript be accepted for publication, we will hold it until you provide the relevant accession numbers or DOIs necessary to access your data. If you wish to make changes to your Data Availability statement, please describe these changes in your cover letter and we will update your Data Availability statement to reflect the information you provide

Reviewers' comments:

Reviewer's Responses to Questions

**Comments to the Author**

1. Is the manuscript technically sound, and do the data support the conclusions?

Reviewer #1: Yes

Reviewer #2: No

2. Has the statistical analysis been performed appropriately and rigorously? 

Reviewer #1: Yes

Reviewer #2: No

3. Have the authors made all data underlying the findings in their manuscript fully available?

Reviewer #1: Yes

Reviewer #2: No

4. Is the manuscript presented in an intelligible fashion and written in standard English?

Reviewer #1: Yes

Reviewer #2: Yes

5. Review Comments to the Author

Reviewer #1: Clear and well-written manuscript. The introduction is relevant and theory based. The methods are generally appropriate. The results are clear. The authors make a systematic contribution to the research literature in this area of investigation.

Reviewer #2: This paper describes a new method called MPASS to compare whole proteomes of shotgun metagenomic sequencing data. However, the methodology is not bioinformatically adequate and tool code is not provided. The reviewer doesn't think this paper is a scientifically sound bioinformatics paper.

Representative points of the authors must be reconsidered are as follows.

Line 383:

When calculating const1 and const2, the reviewer was surprised that the authors do not conduct any correction for multiple substitutions in the same site. Since uncorrected methods tend to underestimate genetic distance, especially for the comparison between distantly related taxa (See molecular evolution textbook), the authors must correct multiple substitutions. Estimating const1 and const2 without correcting multiple substitutions is nonsense.

Line 402:

Ref. 26 does not say these taxa inhabit soils. Taxa choices are not adequate.

To my knowledge, some of them do not inhabit soils.

Sulfolobus islandicus: thermophilic archaea, habitat is hot springs

Proteus mirabilis: pathogen, habitat is mammal gut but can be found in polluted soil

Nitrosospira multiformis: habitat is soil. OK.

Bacteroides fragilis: a very famous human gut microbe

Acidobacterium capsulatum: habitat is soil. OK.

Figure S1:

This figure does not indicate substitution rates but just sequence identities.

Provide the program code for MPASS. Otherwise other researchers cannot use MPASS. Maybe MPASS is a pipeline, so provide complete code is difficult. But at least the authors should provide some shell scripts that are merged some steps of MPASS processes in Fig. 1.

These comments are only representative of methodologies. The reviewer has many other comments related to the metagenomic analysis results of this paper, but before the analysis results, using adequate methodology is necessary.

6. PLOS authors have the option to publish the peer review history of their article (what does this mean?). If published, this will include your full peer review and any attached files.

Reviewer #1: No

Reviewer #2: No

---

## [Author Response · Author response to Decision Letter 0]

8 Jan 2023

Dr. Alfonso Esposito

Academic Editor

PLOS ONE

Dear Dr. Alfonso Esposito,

We would like to thank you and the reviewers for your handling our manuscript and also for the valuable comments. We agreed with most of the comments by you and all reviewers and extensively revised our manuscript. Responses to the “Journal requirements” and the reviewer’s comments are indicated below and in the attached MS word file “Response to Reviewers_SSatoh.docx”. The parts of the attached manuscript that have been revised are highlighted in red (RevisedMS_Track_Changes_SSatoh.docx).

According to the reviewer’s comments, we improved our methodology by the application of a new calculation method, and provided the merged program code for almost all of the steps in MPASS processes and the additional Perl script for the automatic prefiltering of fastq files. We reconstructed all metagenomic trees and their related data with the improved program. We also corrected the sentences including the phrase “data not shown” in the manuscript, and provided the minimal data set for the results described in the manuscript according to the journal requirements of PLOS ONE. Although almost all reconstructed metagenomic trees and their related data were identical to those of the previous version of the manuscript, minor corrections associated with these reconstructed data in the manuscript are highlighted in red. We also highlighted the slight descriptive errors and correction of the format of the manuscript in red. I hope that these revisions are sufficient to make our manuscript suitable for publication in PLOS ONE.

Sincerely yours,

Soichirou Satoh

[Journal requirements]

First requirement; 

Second requirement; 

Answer to the first requirement; 

Line 323 in Page 19 (Discussion):

We deleted the phrase “data not shown” and added the following sentence; 

The topology of the metagenomic trees of simulated metagenomic datasets (Figs. 2 and 3), soil (Fig. 4), and aquatic samples (Fig.5) detected reliable group relationships, however, the integration of the coverage of each gene into metagenomic distances will require further theoretical consideration in future studies.

Line 374 in Page 22 (Materials and Methods):

We deleted the following sentence;

The removal of short protein sequences contributed to the reconstruction of congruent trees (data not shown).

Answer to the second requirement; 

We attached the original data for the metaphylogenomic trees and their related analyses as supporting information files (S1File.zip).

 

[Answers to the comment of Reviewer 2]

Thank you for your critical and careful reading. These comments are valuable for improving our manuscript and I greatly appreciate them. We have revised our manuscript according to your suggestions. I hope you will satisfy our revision. The followings are the answers to your comments.

Comment; 

Line 383:

When calculating const1 and const2, the reviewer was surprised that the authors do not conduct any correction for multiple substitutions in the same site. Since uncorrected methods tend to underestimate genetic distance, especially for the comparison between distantly related taxa (See molecular evolution textbook), the authors must correct multiple substitutions. Estimating const1 and const2 without correcting multiple substitutions is nonsense.

Answer to the comment; 

We fully agreed with your comment that correction for multiple substitutions in the same site is very important. As this suggestion, we improved our methodology by adopting the Poisson-corrected substitution rate for the calculation of the metagenomic distances, and reconstructed all metagenomic trees and their related data with this new calculation method. Although almost all reconstructed metagenomic trees reproduced the same topology as that of trees constructed with the previous MPASS program, we think that our method is improved by your suggestions.

Comment; 

Line 402:

Ref. 26 does not say these taxa inhabit soils. Taxa choices are not adequate.

To my knowledge, some of them do not inhabit soils.

Sulfolobus islandicus: thermophilic archaea, habitat is hot springs

Proteus mirabilis: pathogen, habitat is mammal gut but can be found in polluted soil

Nitrosospira multiformis: habitat is soil. OK.

Bacteroides fragilis: a very famous human gut microbe

Acidobacterium capsulatum: habitat is soil. OK.

Answer to the comment; 

We replaced the phrase “five soil bacteria species” with “five bacteria species”.

Comment;

Figure S1:

This figure does not indicate substitution rates but just sequence identities.

Answer to the comment; 

We calculated the Poisson-corrected substitution rates between 16S rRNA genes of various organisms, and showed the relationship of them to distances from the comparison of metagenomes in Fig. S1.

Comment;

Provide the program code for MPASS. Otherwise other researchers cannot use MPASS. Maybe MPASS is a pipeline, so provide complete code is difficult. But at least the authors should provide some shell scripts that are merged some steps of MPASS processes in Fig. 1. These comments are only representative of methodologies. The reviewer has many other comments related to the metagenomic analysis results of this paper, but before the analysis results, using adequate methodology is necessary.

Answer to the comment; 

We fully agreed with your comment that the availability of the MPASS program by other researchers is also very important. I also hope that this program is used by many researchers to study various metagenomes. According to this suggestion, we provided two Perl scripts. First, we provided automatic program code for the down-sampling, assembling, gene prediction, and calculation of the metagenomic distances in the MPASS pipeline. Second, we also provided another Perl script for the automatic prefiltering of multiple pairs of fastq files by using the Trimmomatic software. We uploaded these two scripts (MPASS.pl, FilteringFastq_for_MPASS.pl) and a refined version of the previous script MPASS_core.pl (only for the metagenomic distance calculation) on the GitHub website (https://github.com/s0sat/MPASS), and wrote this URL in the manuscript. By using MPASS.pl and FilteringFastq_for_MPASS.pl, the researchers can construct the metagenomic distance matrices from their paired-end fastq files or those downloaded from public databases, automatically.

---

## [Editor Report · Decision Letter 1]

20 Jan 2023

Phylogeny analysis of whole protein-coding genes in metagenomic data detected an environmental gradient for the microbiota

PONE-D-22-20665R1

Dear Dr. Satoh,

We’re pleased to inform you that your manuscript has been judged scientifically suitable for publication and will be formally accepted for publication once it meets all outstanding technical requirements.

Kind regards,

Alfonso Esposito, PhD

Academic Editor

PLOS ONE
---

## [Editor Report · Acceptance letter]

24 Jan 2023

PONE-D-22-20665R1 

Phylogeny analysis of whole protein-coding genes in metagenomic data detected an environmental gradient for the microbiota 

Dear Dr. Satoh:

I'm pleased to inform you that your manuscript has been deemed suitable for publication in PLOS ONE. Congratulations! Your manuscript is now with our production department. 

Kind regards, 

on behalf of

Dr. Alfonso Esposito 

Academic Editor

PLOS ONE